# A Novel Graph-based Fuzzy Clustering for Deep Visual Representations

## Abstract

Clustering has long been prized for its simplicity and efficiency. However, with the rapid progress of large-scale self-supervised models, this landscape has shifted. While stronger representations substantially benefit clustering, jointly learning representations and clusters on neural networks has become increasingly resource-intensive. This creates a gap: traditional clustering algorithms remain lightweight but struggle with deep representations, whereas deep clustering methods are effective but computationally expensive. To bridge this gap, we propose Graph-based Fuzzy Clustering (GFC), a novel algorithm to collaborate with foundation models for direct representation clustering. GFC unifies a global fuzzy clustering objective with a local consistency constraint, enabling it to capture both global structural dependencies and local intrinsic connectivity. Furthermore, we develop a fast optimization program to efficiently solve the proposed multi-constrained problem. Extensive experiments across multiple benchmarks demonstrate that GFC not only surpasses state-of-the-art deep clustering methods in accuracy but also achieves superior efficiency, offering a simple and scalable solution for modern deep representations, with a notable 73.4% clustering accuracy on ImageNet-1k.

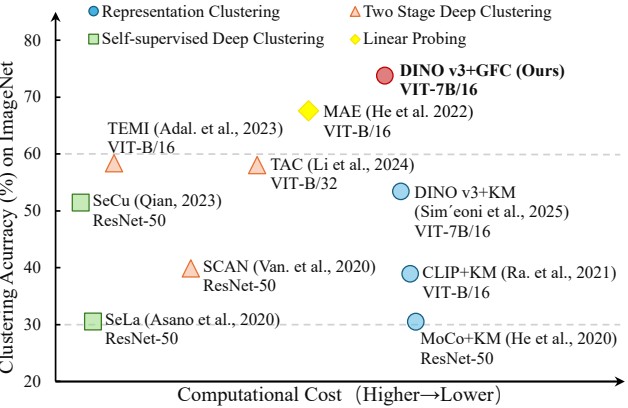

Figure 1: Clustering performance of various methods on ImageNet-1k. We categorize these image clustering approaches into three families: (1) self-supervised deep clustering, which trains a clustering network from scratch using self-supervised learning; (2) two-stage deep clustering, which clusters features from a pre-trained model with additional clustering fine-tuning; (3) representation clustering, which directly clusters the representations produced by a foundation model without using any neural networks. Our proposed GFC achieves a strong balance between efficiency and accuracy, reaching 73.3% clustering accuracy on ImageNet based on DINO v3 representations, which outperforms supervised linear probing result of MAE (He et al., 2022).

## 1 Introduction

Clustering is expected to be lightweight, efficient, and broadly applicable. Recent advances in self-supervised representation learning have shifted this landscape. Stronger representations benefit clus-

tering (Li et al., 2020; Adal. et al., 2023), but obtaining them typically requires scaling up neural networks (Zhai et al., 2022), which makes the joint optimization of representation learning and clustering demanding substantial computational resources. Many deep clustering methods (Li et al., 2022b) further rely on large batch size to stabilize training, which amplifies computational costs. This trend conflicts with the long-standing principle of simplicity in clustering, raising the question: can we design a clustering method that remains efficient and low-cost, buy scales effectively to modern deep representations?

Conventional clustering methods (Liu et al., 2022; Tang et al., 2023; Nie et al., 2023) can be directly applied to the output representations of large models, directly reducing computational complexity. However, most of these algorithms are designed for relatively simple datasets, such as discrete points or low-dimensional manifolds. In contrast, representations extracted from foundation models are typically dense and approximately Gaussian (Wang & Isola, 2020), which causes conventional clustering approaches to encounter severe performance bottlenecks. From a data processing perspective, these conventional clustering methods can be broadly categorized into two families: local clustering, which emphasizes neighborhood consistency, and global clustering, which optimizes a holistic optimization objective. Local clustering methods generally rely on heuristic procedures rather than explicit objective functions, emphasizing fine-grained interactions between instances to yield detailed data partitions. For instance, First Integer Neighbor Clustering Hierarchy (FINCH) (Sarfraz et al., 2019) merges the nearest neighbors in an agglomerative manner. However, local clustering present practical challenges when processing deep representations. The dense and high-dimensional nature of deep features makes these methods often fail to produce meaningful clusters. Moreover, their theoretical simplicity comes at the cost of requiring additional hyperparameter tuning (Schubert et al., 2017) and incurring high computational overhead (Murtagh & Contreras, 2012; Frey & Dueck, 2007). Global clustering methods aim to group data by optimizing a specific objective function. K-means and its variants (Yang et al., 2017; Nie et al., 2021b; 2022) are among the most widely adopted algorithms. Nevertheless, when applied to deep representations, these methods face significant limitations. They often fail to preserve fine-grained neighborhood relationships, which reduces their effectiveness for detailed representation hyperplane partitioning (Xie et al., 2016). By contrasting local and global approaches, it is evident that each paradigm captures only part of the data characteristics, making it difficult to fully exploit the rich and multi-scale structure of deep representations.

In this paper, we propose a novel Graph-based Fuzzy Clustering (GFC) algorithm, which integrates global and local clustering to overcome the limitations in representation clustering of deep neural networks. GFC integrates a global objective for coarse-grained distribution clustering with a local neighborhood constraint for fine-grained structural learning, enabling it to effectively adapt to the dense and Gaussian-like distribution characteristics of deep representations. GFC aggregates graph-based knowledge in a fuzzy manner, effectively capturing both local intrinsic connectivity and global structural dependencies. This formulation results in a multi-constrained quadratic optimization problem. To solve it efficiently, we incorporate a fuzzy weighting exponent and a hard assignment matrix into the objective function. The fuzzy weighting exponent controls the degree of fuzziness in the probability outputs, while the hard assignment matrix serves as an auxiliary variable to accelerate convergence. And a concise optimization program is proposed to solve the problem iteratively. Experimental results across multiple benchmarks demonstrate that GFC achieves state-of-the-art performance in clustering deep visual representations. Notably, as shown in Fig. 1, GFC attains 73% clustering accuracy on ImageNet using the DINO v3 foundation model.

- We propose a clustering model with a well-defined objective function based on the sum of cluster co-occurrence probabilities and a local clustering constraint to address the difficulty of clustering deep representations.

- We propose a Lagrangian optimization algorithm and an acceleration scheme to efficiently solve the proposed model. The proposed algorithm ensures faster convergence and enhances the scalability of the clustering model, making it well-suited for large-scale data.

- Extensive experiments have been conducted on widely used image benchmarks, validating that our method yiels satisfactory performance on both accuracy and efficiency.

## 2 RELATED WORK

**Fuzzy and Graph Clustering.** Fuzzy C-Means (FCM) (Bezdek et al., 1984) is a classical fuzzy clustering algorithm. Many variants have been developed (Krishnapuram & Keller, 1993; Nie et al., 2021a) to improve its adaptability and performance. For instance, Probability Aggregation Clustering (PAC) (Yan et al., 2024) eliminates explicit cluster centers to achieve a more robust optimization. For graph-based clustering, spectral clustering (SC) (Von Luxburg, 2007) and its variants are most widely used algorithms. Self-Constrained Spectral Clustering (Self-CSC) (Bai et al., 2022) enhances SC by incorporating pairwise and label self-constrained terms, enabling high-quality clustering without prior information. Despite these advances, traditional fuzzy and graph-based methods are primarily designed for toy datasets and struggle with deep neural network outputs, facing challenges in fine-grained partitioning, parameter tuning, and algorithmic efficiency.

**Deep Clustering.** Learning representations while performing clustering has achieved notable success in natural image clustering. Contrastive Clustering (Li et al., 2021) jointly optimizes feature and clustering heads to perform semantic clustering. In addition, vision-language pre-training (VLP) methods (Radford et al., 2021; Li et al., 2022a) enhance deep image clustering by leveraging text representations as guidance (Cai et al., 2023; Li et al., 2024). However, deep clustering still faces challenges. For example, joint representation learning and clustering is computationally expensive (Yan et al., 2024), and scaling to high-capacity models increases memory and time costs (Zhai et al., 2022). These issues motivate the need for efficient representation clustering methods.

## 3 PRELIMINARIES

$\mathcal{X} = \{\boldsymbol{x}_i\}_{i=1}^n$ denotes an $n$-point set, where $\boldsymbol{x}_i \in \mathbb{R}^d$ is the $i^{th}$ $d$-dimensional sample. $\boldsymbol{W}$ denotes the similarity graph of $\mathcal{X}$. Our work aims to study a cluster probability matrix $\boldsymbol{P} = [p_{i,j}]_{n \times c}$, which satisfies $\boldsymbol{P} \in \mathcal{P} = \{[p_{i,l}]_{n \times c}|p_{i,l} \in [0,1]; \sum_{l=1}^c p_{i,l} = 1; 0 < \sum_{i=1}^n p_{i,l} < n, i = 1, \cdots, n, l = 1, \cdots, c\}$. First, we introduce a cluster co-occurrence probability $\rho_{i,j}$ that can be represented as

$$\rho_{i,j} = \boldsymbol{p}_i^T \boldsymbol{p}_j = \sum_{l=1}^c p_{i,l} p_{j,l}, \tag{1}$$

where $\rho_{i,j} \in [0,1]$. In Eq. (1), the inner product of two probability vectors measures the probability that the two samples belong to the same cluster. Therefore, an intuitive clustering approach is to maximize $\rho_{\cdot,\cdot}$ between neighbors. And the optimization problem is formulated as

$$\min_{\boldsymbol{P} \in \mathcal{P}} \sum_{i=1}^n \sum_{j \neq i} \rho_{i,j} - \alpha \sum_{i=1}^n \sum_{j \in \mathcal{A}_i} \rho_{i,j} w_{i,j}, \tag{2}$$

where $\alpha > 0$ is a penalty hyperparameter. And Eq. (2) can be expressed in matrix form

$$\min_{\boldsymbol{P} \in \mathcal{P}} Tr(\boldsymbol{P}^T(\boldsymbol{H} - \alpha \boldsymbol{W})\boldsymbol{P}), \tag{3}$$

where $\boldsymbol{H} = \mathbf{1}_n^T \mathbf{1}_n - \boldsymbol{I}$, $\mathbf{1}_n$ is the 1 vector of length $n$. The first term of Eq. (2) is the self-constrained term, acting as a regularization term to penalize trivial solutions. The second term of Eq. (2) is the clustering term that promotes the coherence of the clustering predictions across the overall data distribution. It encourages the algoritm to assign similar samples to the same cluster.

## 4 CLUSTERING BY GLOBAL AND LOCAL PERSPECTIVE

### 4.1 GLOBAL CLUSTERING OBJECTIVE FUNCTION

Based on the conception of Eq. (3), the objective function of GFC is defined as

$$Tr(\boldsymbol{V}^T \boldsymbol{H} \boldsymbol{V} + \boldsymbol{P}^T \boldsymbol{H} \boldsymbol{P}^m - \alpha \boldsymbol{V}^T \hat{\boldsymbol{W}} \boldsymbol{P}^m), \tag{4}$$

where $\boldsymbol{V}$ is the hard assignment matrix that satisfies $\boldsymbol{V} \in \mathcal{V} = \{[v_{i,l}]_{n \times c}|v_{i,l} \in \{0,1\}; \sum_{l=1}^c v_{i,l} = 1; 0 < \sum_{i=1}^n v_{i,l} < n, i = 1, \cdots, n, l = 1, \cdots, c\}$. $m \in (1, +\infty)$ and $\boldsymbol{P}^m = [p_{i,k}^m] \in \mathbb{R}^{+n \times c}$

denotes that every element of the matrix is raised to a power. $\boldsymbol{L}$ is the Laplacian matrix of graph $\boldsymbol{W}$. In Eq. (4), the first two terms are self-constraint terms, while the third term corresponds to the clustering term. We introduce $\boldsymbol{V}$ as an auxiliary variable consisting of 0 and 1 to facilitate optimization. $\boldsymbol{V}$ accelerates convergence in the early stages of optimization, making $\boldsymbol{P}$ more confident. In later stages, $\boldsymbol{V}$ is updated less frequently than $\boldsymbol{P}$, which improves the stability of the model.

## 4.2 LOCAL CLUSTERING CONSTRAINT

We incorporate a local consistency constraint into the model to make the model better able to handle complex and dense representation data. The constraint is defined as

$$\boldsymbol{LP} = (\boldsymbol{\Delta} - \boldsymbol{W})\boldsymbol{P} = 0, \tag{5}$$

where $\boldsymbol{L} = \boldsymbol{\Delta} - \boldsymbol{W}$ is the Laplacian matrix, $\boldsymbol{\Delta}$ is the degree matrix of $\boldsymbol{W}$, $\Delta_{i,i} = \sum_{j=1}^{n} w_{i,j}$. Right multiply both sides of the equation by matrix $\boldsymbol{\Delta}^{-1}$, we will get $\boldsymbol{P} = \boldsymbol{\Delta}^{-1}\boldsymbol{W}\boldsymbol{P}$, which establishes the relation of predictions between the instance and its $k$-NN.

## 4.3 CLUSTERING PROBLEM

By unifying the global and local clustering definition, the entire GFC optimization problem can be formulated as

$$\min_{\boldsymbol{P} \in \mathcal{P}, \boldsymbol{V} \in \mathcal{V}} Tr(\boldsymbol{V}^T \boldsymbol{H} \boldsymbol{V} + \boldsymbol{P}^T \boldsymbol{H} \boldsymbol{P}^m - \alpha \boldsymbol{V}^T \hat{\boldsymbol{W}} \boldsymbol{P}^m),$$
$$\text{s.t.} \quad \boldsymbol{LP} = 0, \tag{6}$$

The objective function aims to identify a reasonable grouping pattern from the global-view. And the constraint ensures the prediction of sample $i$ is consistent with its local structure from the point-view.

The clustering model involves two graphs: the $k$-NN graph $\boldsymbol{W}$ and the adjacency indicator graph $\hat{\boldsymbol{W}}$. $\boldsymbol{W} = [w_{i,j}]_{n \times n}$ is constructed by

$$w_{i,j} = \begin{cases} \exp{(-\frac{\|\boldsymbol{x}_i - \boldsymbol{x}_j\|_2^2}{2\sigma})}, & j \in \mathcal{N}^k(i) \text{ or } i \in \mathcal{N}^k(j) \\ 0, & \text{otherwise} \end{cases} \tag{7}$$

where $\sigma$ is the kernel parameter, $\mathcal{N}^k(i)$ denotes the set of $k$-nearest neighbors of sample $i$. And the adjacency indicator $\hat{\boldsymbol{W}}$ is calculated by

$$\hat{\boldsymbol{W}} = \mathbb{I}((\boldsymbol{W} + \boldsymbol{I})^\theta > 0) - \boldsymbol{I}, \tag{8}$$

where $\theta$ is a positive integer, $\mathbb{I}(\cdot)$ is the element-wise indictor function, $\mathbb{I}(\boldsymbol{W} > 0) = [\mathbb{I}(w_{i,j} > 0)]_{n \times n}$, which equals one when $w_{i,j} > 0$. $\hat{\boldsymbol{W}}$ is introduced to find enough neighborhood to support clustering due to $k$ being too small. And the hyperparameter $\theta$ controls the size of neighborhood, for which we adopt an empirically setting

$$\theta = \lceil \log_k \frac{n}{c} \rceil, \tag{9}$$

where $\lceil \cdot \rceil$ is the integer up function, so that the neighborhood number of each sample are close to $n/c$.

## 4.4 ALTERNATE OPTIMIZATION FOR GFC

Eq. (6) is a multiconstrained optimization problem. It is noted that $\boldsymbol{P}$ and $\boldsymbol{V}$ can be computed separately with each other fixed, so the objective function of GFC is optimized by alternately updating $V$ and $P$.

**Fix $V$ and Update $P$.** Since the modeling of Eq. (6) is not conducive to direct solution, we treat local clustering constraint as a slack constraint and use the alternating projection optimization algorithm (Escalante & Raydan, 2011) to solve $\boldsymbol{P}$. Specifically, the objective function without local clustering constraint is simplified to

$$\min_{\boldsymbol{P} \in \mathcal{P}} Tr((\boldsymbol{P}^T(\boldsymbol{1}_n \boldsymbol{1}_n^T - \boldsymbol{I}) - \alpha \boldsymbol{V}^T \hat{\boldsymbol{W}})\boldsymbol{P}^m). \tag{10}$$

By using the Lagrangian multiplier method, the problem can be decomposed into $n$ unconstrained sub-problems. By solving the sub-problems, we can finally obtain following equivalence relation

$$p_{i,l}^* = \frac{s_{i,l}^{p}{}^{-1/(m-1)}}{\sum_{r=1}^{c} s_{i,r}^{p}{}^{-1/(m-1)}}, \forall i, l, \tag{11}$$

where $s_{i,l}^p$ is the fuzzy clustering score,

$$s_{i,l}^p = \sum_{j \neq i} (p_{j,l} - \alpha v_{j,l} \hat{w}_{i,j}) \in (0, +\infty), \tag{12}$$

and has vector form $\boldsymbol{s}_i^p = [s_{i,1}^p, \cdots, s_{i,c}^p]^T = (\boldsymbol{P}^T \boldsymbol{1}_n - \boldsymbol{p}_i) - \alpha \boldsymbol{V}^T \hat{\boldsymbol{w}}_i$, where $\hat{\boldsymbol{w}}_i$ is the $i$-th row of matrix $\hat{\boldsymbol{W}}$. The fuzzy clustering score indicates the score that $\boldsymbol{x}_i$ belongs to the cluster $k$.

Then, we project the obtained $\boldsymbol{P}^*$ into the feasible region according to the local clustering constraint. The projected optimization problem is formulated as

$$\min_{\boldsymbol{P} \in \mathcal{P}} \|\boldsymbol{P} - \boldsymbol{P}^*\|_{\mathcal{F}}^2,$$
$$\text{s.t.} (\boldsymbol{I} - \boldsymbol{W} \boldsymbol{\Delta}^{-1}) \boldsymbol{P} = 0. \tag{13}$$

our local clustering constraint is a slack constraint, which doesn't need to be strictly followed. Therefore, we relax the constraint to the objective function and transform Eq. (13) into following problem

$$\min_{\boldsymbol{P} \in \mathcal{P}} \|\boldsymbol{P} - \boldsymbol{P}^*\|_{\mathcal{F}}^2 + \beta \|\boldsymbol{P} - \boldsymbol{W} \boldsymbol{\Delta}^{-1} \boldsymbol{P}\|_{\mathcal{F}}^2, \tag{14}$$

where $\beta > 0$ is the penalty hyperparameter. Eq. (14) is a label propagation problem (Zhou et al., 2003) that limits the feasible domain to the probability space. Afterwards, similar as above solving process, we can get following solution

$$p_{i,l} = \frac{1}{1+\beta} \frac{s_{i,l}^{p}{}^{-1/(m-1)}}{\sum_{r=1}^{c} s_{i,r}^{p}{}^{-1/(m-1)}} + \frac{\beta}{1+\beta} \frac{\sum_{j=1}^{n} w_{i,j} p_{j,l}}{\sum_{j=1}^{n} w_{i,j}}. \tag{15}$$

The first term is the extreme point of the objective function without local clustering constraint, and the second term is the weighted average of the predictions of $k$-NN.

**Fix $\boldsymbol{P}$ and Upadte $\boldsymbol{V}$** Taking $\boldsymbol{v}_i$ as a variable and the others as constant, the problem in Eq. (6) degenerates to

$$\min_{\boldsymbol{v}_i} (\boldsymbol{V}^T \boldsymbol{1}_n - \boldsymbol{v}_i - \alpha \boldsymbol{P}^{mT} \hat{\boldsymbol{w}}_i)^T \boldsymbol{v}_i, \tag{16}$$

Similar to $\boldsymbol{P}$, there is also a hard clustering score $s_{i,l}^v \in \mathbb{R}^+$ $(i = 1 \cdots, n; l = 1 \cdots, c)$

$$\boldsymbol{s}_i^v = [s_{i,1}^v, \cdots, s_{i,c}^v]^T = \boldsymbol{V}^T \boldsymbol{1}_n - \boldsymbol{v}_i - \alpha \boldsymbol{P}^{mT} \hat{\boldsymbol{w}}_i. \tag{17}$$

Based on the properties of the hard assignment matrix, the minimum point of Eq. (16) corresponds to the minimum point of the hard clustering score $\boldsymbol{s}_i^v$ is

$$\begin{cases} v_{i,l} = 1, & l = l^*, \\ v_{i,l} = 0, & l \neq l^*, \end{cases}$$
$$\text{with} \quad l^* = \arg \min_l (\boldsymbol{s}_i^v)_l, \forall i. \tag{18}$$

Finally, according to the deduction, $\boldsymbol{P}$ and $\boldsymbol{V}$ can be solved by iteratively updating Eq. (15) and Eq. (18) for each $\boldsymbol{p}_i$ and $\boldsymbol{v}_i$ until converging.

## 4.5 AGGREGATION ACCELERATION

Based on the theoretical solution of $\boldsymbol{P}$ and $\boldsymbol{V}$, we can alternately solve the GFC optimization problem. However, in practical implementation, the floating-point calculations of the clustering score in Eq. (12) and Eq. (17) consume significant computational resources. For each clustering score, the

calculation requires about $2nc$ times additions and multiplications, resulting in quadratic time complexity for the entire iteration process. Therefore, it is essential to reduce computational complexity to linear to better handle large-scale problems. Let $\boldsymbol{P}(t)$ denote the iterative sequence of $\boldsymbol{P}$ in the optimization program, where $t$ is the iteration index. Eq.(12) and Eq.(17) can be rewritten as

$$\boldsymbol{s}_i^p(t+1) = \tilde{\boldsymbol{P}}(t) - \boldsymbol{p}_i(t) - \alpha \sum_{j \in \mathcal{A}_i} \boldsymbol{v}_j(t), \tag{19}$$

$$\boldsymbol{s}_i^v(t+1) = \tilde{\boldsymbol{V}}(t) - \boldsymbol{v}_i(t) - \alpha \sum_{j \in \mathcal{A}_i} \boldsymbol{p}_j(t)^m, \tag{20}$$

where $\tilde{\boldsymbol{P}}(t) = \sum_{i=1}^n \boldsymbol{p}_i(t)$ and $\tilde{\boldsymbol{V}}(t) = \sum_{i=1}^n \boldsymbol{v}_i(t)$, $\mathcal{A} = \{\mathcal{A}_1, \cdots, \mathcal{A}_n\}$ is the adjacent sample set $\mathcal{A}_i = \{j | w_{i,j} = 1\}$. By Eq. (19) and Eq. (20), most vector multiplication operations are eliminated. Furthermore, $\tilde{\boldsymbol{P}}(t+1)$ and $\tilde{\boldsymbol{V}}(t+1)$ can be updated by

$$\tilde{\boldsymbol{P}}(t+1) = \tilde{\boldsymbol{P}}(t) - \boldsymbol{p}_i(t) + \boldsymbol{p}_i(t+1), \tag{21}$$

$$\tilde{\boldsymbol{V}}(t+1) = \tilde{\boldsymbol{V}}(t) - \boldsymbol{v}_i(t) + \boldsymbol{v}_i(t+1), \tag{22}$$

which denotes that $\tilde{\boldsymbol{P}}$ and $\tilde{\boldsymbol{V}}$ can be reused. Therefore, GFC only needs to add the vector in $\mathcal{A}_i$ in each iteration. Besides, most calculations of clustering scores are redundant, as the numerical relation within $\boldsymbol{s}_i^v$ and $\boldsymbol{s}_i^p$ does not change significantly when computed over subsets of the data. Therefore, it is natural to consider using a mini-batch clustering algorithm to speed up the optimization process. We randomly divide the dataset into $n_b$ mutually disjoint subsets, $\mathcal{X} = \{\mathcal{X}_1, \ldots, \mathcal{X}_{n/n_b}\}$, where $n_b$ is the batch size. Vectors $\boldsymbol{s}_i^v$ and $\boldsymbol{s}_i^p$ are computed in subsets $\mathcal{X}_j$, with $\boldsymbol{x}_i \in \mathcal{X}_j$ to reduce the number of computational elements in $\mathcal{A}_i$. Notably, the proposed local constraint mitigates the potential degradation of mini-batch operations by ensuring that samples remain consistent with their respective neighborhoods.

### 4.6 ALGORITHM DETAILS

#### 4.6.1 HYPERPARAMETERS SELECTION

The main hyperparameters of GFC are the number of neighbors $k$, the fuzzy weighting exponent $m$, the regularization parameter $\alpha$, and the penalty parameter $\beta$. The exponent $m$ controls the sharpness of probability outputs, and values closer to 1 yield more confident assignments. The parameter $\alpha$ controls the strength of the regularization term. The parameter $\beta$ controls the local consistency constraint. We schedule it to increase along with the iterative steps. In this paper, results without explicitly stated were obtained under the same settings of $k = 20, m = 1.05, \alpha = 0.5$. Additional details, including Pseudocode and parameter Sensibility Analysis, are provided in the Appendix 8.

#### 4.6.2 TIME COMPLEXITY

By introducing aggregation acceleration strategy, GFC requires approximately $n_b$ addition operations to compute $\boldsymbol{s}_i^p$ and $\boldsymbol{s}_i^v$. Hence, the time complexity of the core GFC optimization program is $\mathcal{O}(n_b n)$, which is much lower than that of SGD in the deep clustering methods.

## 5 EXPERIMENT

We compare GFC with several baselines across multiple benchmarks to demonstrate its superiority. Beyond quantitative comparisons, we further conduct qualitative studies to validate the effectiveness of individual components and to provide deeper insights into the behavior of GFC.

### 5.1 DEEP REPRESENTATION ACQUISITION

With the rise of large-scale pre-training, high-quality image representations can be obtained from various pre-trained models. We adopt three representative visual representation extractors to evaluate GFC: SimCLR (Chen et al., 2020b), CLIP (Radford et al., 2021), and DINO v3 (Siméoni et al., 2025). Specifically, we use ResNet-34, ViT-B/16, and ViT-7B/16 pre-trained by SimCLR, CLIP, and DINO v3, respectively, as feature extractors. The resulting representations serve as inputs for

Table 1: Dataset settings for our experiments. On CIFAR-10, CIFAR-20, and STL-10, features extracted from SimCLR and CLIP are 512-d, whereas features from DINO v3 are 4096-d.

| Dataset | Sample | Cluster | Dimension | Data Type |
|---|---|---|---|---|
| Pendigits (Alimoglu & Alpaydin, 1997) | 10,992 | 10 | 16 | atifical feature |
| Mnist (Deng, 2012) | 70,000 | 10 | 784 | raw image data |
| USPS (Hull, 1994) | 9,298 | 10 | 256 | raw image data |
| CIFAR-10 (Krizhevsky et al., 2009) | 60,000 | 10 | 512/4096 | deep feature |
| CIFAR-20 (Krizhevsky et al., 2009) | 60,000 | 20 | 512/4096 | deep feature |
| STL-10 (Coates et al., 2011) | 13,000 | 10 | 512/4096 | deep feature |
| ImageNet (Deng et al., 2009) | 1,281,167 | 1,000 | 4096 | deep feature |

Table 2: Performance of different algorithms on deep representations. The results are organized into three categories according to the self-supervised backbone: SimCLR, CLIP, and DINO v3. Results for traditional clustering methods are reported as the Avg ± Std over 50 independent runs, while for deep learning methods we present the best-performing result.

| Dataset | | CIFAR-10 | | | CIFAR-20 | | | STL-10 | | |
|---|---|---|---|---|---|---|---|---|---|---|
| Metric | | NMI | ACC | ARI | NMI | ACC | ARI | NMI | ACC | ARI |
| Representation pre-training method: SimCLR (Chen et al., 2020a); Architecture: ResNet-34; Parameter: ∼22M | | | | | | | | | | |
| TCC(Shen et al., 2021) | Best | 79.0 | 90.6 | 73.3 | 47.9 | 49.1 | 31.2 | 73.2 | 81.4 | 68.9 |
| SPICE(Niu et al., 2022) | Best | 86.5 | 92.6 | 85.2 | 56.7 | 53.8 | 38.7 | 87.2 | 93.8 | 87.0 |
| TCL(Li et al., 2022b) | Best | 81.9 | 88.7 | 78.0 | 52.9 | 53.1 | 35.7 | 79.9 | 86.8 | 75.7 |
| ProPos (Huang et al., 2022) | Best | 88.6 | 94.3 | 88.4 | 60.6 | 61.4 | 45.1 | 75.8 | 86.7 | 73.7 |
| DPAC (Yan et al., 2024) | Best | 87.0 | 93.4 | 86.6 | 54.2 | 55.5 | 39.3 | 86.3 | 93.4 | 86.1 |
| UCD (Chang et al., 2025) | Best | 88.1 | 94.3 | 88.7 | 58.4 | 59.7 | 42.9 | 89.5 | 91.9 | 89.1 |
| KM + SimCLR | Avg | 76.4 | 80.9 | 67.4 | 47.7 | 43.7 | 26.5 | 66.9 | 69.0 | 52.0 |
| | ±Std | ±0.8 | ±2.4 | ±1.1 | ±1.0 | ±1.2 | ±1.0 | ±0.8 | ±0.9 | ±1.0 |
| **GFC + SimCLR** | Avg | 83.2 | 90.4 | 81.4 | 51.4 | 48.6 | 33.5 | 77.9 | 87.2 | 74.9 |
| | ±Std | ±0.0 | ±0.0 | ±0.0 | ±0.8 | ±1.3 | ±1.0 | ±0.0 | ±0.0 | ±0.1 |
| Representation pre-training method: CLIP (Radford et al., 2021); Architecture: VIT-B; Parameter: ∼86M | | | | | | | | | | |
| TEMI (Adal. et al., 2023) | Best | 88.6 | 94.5 | 88.5 | 65.4 | 63.2 | 48.9 | 96.5 | 98.5 | 96.8 |
| SIC (Cai et al., 2023) | Best | 84.8 | 92.7 | 84.6 | 59.3 | 58.4 | 44.0 | 95.4 | 98.1 | 95.9 |
| TAC (Li et al., 2024) | Best | 83.3 | 91.9 | 83.1 | 61.1 | 60.7 | 44.8 | 95.5 | 98.2 | 96.1 |
| MCA (Qiu et al., 2024) | Best | 85.0 | 92.8 | 84.9 | 60.6 | 61.2 | 45.5 | 95.5 | 98.2 | 96.0 |
| ECC (Li et al., 2025) | Best | 81.8 | 88.1 | 77.8 | 58.6 | 56.2 | 41.3 | 76.6 | 81.3 | 71.3 |
| CLIP (zero shot) | Avg | 80.7 | 90.0 | 79.3 | 55.3 | 58.3 | 39.8 | 93.9 | 97.1 | 93.7 |
| | ±Std | ±0.0 | ±0.0 | ±0.0 | ±0.0 | ±0.0 | ±0.0 | ±0.0 | ±0.0 | ±0.0 |
| KM + CLIP | Avg | 76.0 | 78.3 | 69.0 | 51.2 | 48.7 | 32.5 | 94.3 | 96.0 | 92.4 |
| | ±Std | ±0.1 | ±0.2 | ±0.2 | ±0.4 | ±0.8 | ±0.5 | ±1.2 | ±1.6 | ±2.8 |
| **GFC + CLIP** | Avg | 86.0 | 92.9 | 86.0 | 61.1 | 59.8 | 44.3 | **97.2** | **98.9** | **97.6** |
| | ±Std | ±0.0 | ±0.0 | ±0.0 | ±0.7 | ±1.6 | ±1.3 | ±0.0 | ±0.0 | ±0.0 |
| Representation pre-training method: DINO v3 (Siméoni et al., 2025); Architecture: VIT-7B; Parameter: ∼7B | | | | | | | | | | |
| KM + DINO v3 | Avg | 88.1 | 80.1 | 75.7 | 58.7 | 44.3 | 30.6 | 59.4 | 47.6 | 25.1 |
| | ±Std | ±2.7 | ±4.6 | ±6.2 | ±4.1 | ±5.6 | ±5.0 | ±5.3 | ±2.6 | ±6.9 |
| **GFC + DINO v3** | Avg | **98.1** | **99.3** | **98.4** | **72.4** | **63.8** | **52.4** | 96.9 | 98.0 | 96.4 |
| | ±Std | ±0.0 | ±0.0 | ±0.0 | ±2.2 | ±3.5 | ±3.2 | ±2.3 | ±2.4 | ±3.8 |

GFC. The representation dimensions are 512 for SimCLR and CLIP, and 4096 for DINO v3. All processes rely on the official models and only involve forward propagation. Therefore, the required GPU memory is very little, ensuring the efficiency of the image clustering model.

## 5.2 DATASET

We evaluate GFC on seven challenging datasets from diverse domains, including image data (MNIST, USPS), artificial feature data (Pendigits), and deep visual representation data (CIFAR-10, CIFAR-20, STL-10, ImageNet). The dataset statistics are summarized in Table 1. Notably, CIFAR-20 is derived from CIFAR-100 by grouping 100 fine-grained classes into 20 superclasses, which is widely used for deep image clustering.

Table 3: Performance of different algorithms on raw data.

| Dataset | | Pendigits | | | USPS | | | Mnist | | |
|---|---|---|---|---|---|---|---|---|---|---|
| Metric | | NMI | ACC | ARI | NMI | ACC | ARI | NMI | ACC | ARI |
| KM (Wang et al., 2014) | Avg | 68.0 | 69.9 | 55.3 | 60.7 | 64.3 | 52.4 | 49.7 | 54.6 | 37.6 |
| | ±Std | ±1.1 | ±5.1 | ±3.4 | ±0.4 | ±2.1 | ±0.9 | ±1.4 | ±2.9 | ±2.0 |
| FCM (Bezdek et al., 1984) | Avg | 68.4 | 68.3 | 56.1 | 62.3 | 63.9 | 53.2 | 48.9 | 54.6 | 37.0 |
| | ±Std | ±1.3 | ±1.3 | ±3.8 | ±1.3 | ±2.2 | ±2.2 | ±1.1 | ±1.7 | ±1.3 |
| SC (Von Luxburg, 2007) | Avg | 72.9 | 71.4 | 55.9 | 48.9 | 56.1 | 29.5 | 47.0 | 55.5 | 36.5 |
| | ±Std | ±1.0 | ±1.4 | ±1.3 | ±0.2 | ±0.2 | ±0.4 | ±0.6 | ±1.6 | ±1.4 |
| FINCH (Sarfraz et al., 2019) | Avg | 72.5 | 62.7 | 50.1 | 77.7 | 67.3 | 61.7 | 72.6 | 62.3 | 54.8 |
| | ±Std | ±0.0 | ±0.0 | ±0.0 | ±0.0 | ±0.0 | ±0.0 | ±0.0 | ±0.0 | ±0.0 |
| K-sums (Pei et al., 2022) | Avg | 65.7 | 72.6 | 57.1 | 63.8 | 71.8 | 57.7 | 50.9 | 58.5 | 42.0 |
| | ±Std | ±0.0 | ±0.0 | ±0.0 | ±0.5 | ±0.8 | ±0.6 | ±1.0 | ±1.3 | ±1.4 |
| Self-CSC (Bai et al., 2022) | Avg | 82.3 | 83.7 | 72.3 | 72.6 | 75.4 | 61.4 | 80.7 | 85.1 | 75.5 |
| | ±Std | ±2.6 | ±3.7 | ±4.6 | ±2.1 | ±3.4 | ±4.3 | ±1.6 | ±3.6 | ±4.0 |
| PAC (Yan et al., 2024) | Avg | 70.2 | 77.3 | 61.6 | 73.4 | 61.7 | 53.0 | 49.5 | 60.7 | 40.9 |
| | ±Std | ±0.4 | ±1.2 | ±0.4 | ±0.4 | ±0.1 | ±0.2 | ±0.5 | ±0.4 | ±0.5 |
| **GFC** | Avg | **83.7** | **86.9** | **77.0** | **83.2** | **81.5** | **77.0** | **85.3** | **87.6** | **82.3** |
| | ±Std) | ±1.6 | ±1.7 | ±2.3 | ±0.2 | ±0.2 | ±0.3 | ±0.1 | ±0.0 | ±0.1 |

Table 4: Comparison of GFC with baselines on large-scale image datasets. SCAN and ProPos adopt ResNet-50 with contrastive learning, and TAC and TEMI adopt ViT-Base with CLIP.

| Dataset | | ImageNet | | | | | | | |
|---|---|---|---|---|---|---|---|---|---|
| Sub-classes | | 100 | | | 200 | | | 1000 | |
| Metric | NMI | ACC | ARI | NMI | ACC | ARI | NMI | ACC | ARI |
| SCAN (Van. et al., 2020) | 80.8 | 68.9 | 57.6 | 77.2 | 58.1 | 47.0 | 74.7 | 44.7 | 32.4 |
| ProPos (Huang et al., 2022) | 83.5 | - | 63.5 | 80.6 | - | 53.8 | - | - | - |
| TEMI (Adal. et al., 2023) | 85.7 | 75.1 | 65.5 | 85.2 | 73.1 | 62.1 | - | 58.4 | 45.9 |
| TAC (Li et al., 2024) | - | - | - | - | - | - | 79.9 | 58.2 | 43.5 |
| GFC + DINO v3 | **92.8** | **92.9** | **89.9** | **90.9** | **87.0** | **80.8** | **87.3** | **73.3** | **63.0** |

## 5.3 MAIN RESULTS

To demonstrate the superiority of GFC in handling deep visual representations, we compare it with several state-of-the-art deep clustering methods on CIFAR-10, CIFAR-20, and STL-10, with results summarized in Table 2. First, deep embeddings are typically dense and highly entangled, which often leads to severely degraded performance for traditional clustering methods such as K-means (KM). Compared with KM, GFC achieves an average improvement of nearly 15% across representations extracted by SimCLR, CLIP, and DINO v3. Notably, results on DINO v3 indicate that GFC is particularly well-suited for clustering large-scale vision models, whose outputs exhibit greater nonlinearity. Second, when compared with deep clustering methods, we observe that as base model capacity and representation quality increase, the performance of GFC shifts from being weaker (on SimCLR) to surpassing these methods (on CLIP). In addition, GFC outperforms CLIP zero-shot predictions, which perform clustering via text labels. GFC combined with DINO v3 achieves substantial improvements over deep clustering methods on CIFAR-10, CIFAR-20, and STL-10. These findings highlight that, beyond the traditional deep clustering paradigm, GFC has the potential to serve as an effective and general solution for large-scale vision pre-trained models. Third, in terms of efficiency, GFC + DINO v3 achieves competitive results with only a few hours to forward pass ViT-7B/16 for representation extraction and a few minutes for clustering. Overall, compared to deep clustering methods, GFC provides notable advantages, including fewer hyperparameters, stronger generality, and more cost-effective clustering.

Table 3 presents the clustering performance of GFC compared to several classical methods on three real-world datasets. According to our definition, KM, FCM, SC, K-sums, and Self-CSC are global clustering methods, while FINCH is a local clustering method. Across all metrics (NMI, ACC, ARI), GFC consistently achieves the best results. On Pendigits, GFC outperforms the second-best method, Self-CSC, by 1.4–4.7 points. But the complexity of self-CSC is cubic, which makes it impossible to process large-scale data. IN summary, these results highlight the effectiveness and robustness of GFC for diverse traditional clustering datasets.

## 5.4 IMAGENET EVALUATION

To evaluate the scalability of GFC on large-scale datasets with more instances and clusters, we conduct experiments on ImageNet. We adopt DINO v3 with the ViT-7B/16 backbone to extract deep features, providing a challenge testbed for evaluating GFC. Following SCAN (Van. et al., 2020), we select subsets of ImageNet with 100, 200, and 1000 classes. To address the memory overhead of the 1.28 million $\times$ 4096 feature matrix on ImageNet-1k, we divide the data into five overlapping blocks, perform clustering within each block, and then compute the final evaluations by aggregating the results from blocks. We evaluate GFC against four deep clustering baselines, reporting the best performance over 5 runs. As shown in Table 4, GFC consistently outperforms other methodsby a significant margin, demonstrating its potential for effective scaling in deep learning.

## 5.5 EFFECT OF AUXILIARY HARD ASSIGNMENT MATRIX

We study the role of the hard assignment matrix $V$, which is designed to accelerate convergence during optimization. We replace $V$ with $P$ and denote the variant as GFC w/o $V$. As shown in Fig. 2, we track accuracy across iterations. The results show that GFC with $V$ notably speeds up convergence in the early stage and stabilizes predictions in the later stage, leading to improved overall performance. We attribute this effect to $V$: in the early stage, the one-shot vector of $V$ provides a large gradient-oriented direction; in the later stage, its slow change stabilizes convergence.

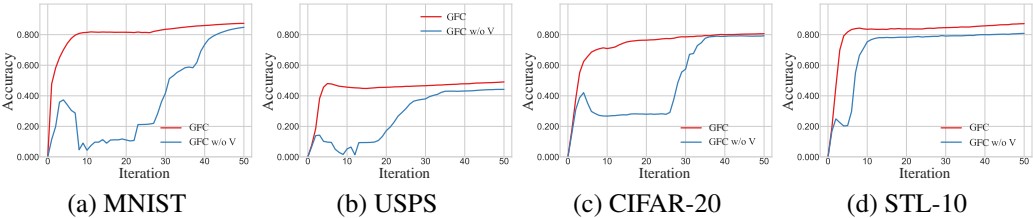

| (a) MNIST | (b) USPS | (c) CIFAR-20 | (d) STL-10 |

Figure 2: Effect of Auxiliary Hard Assignment Matrix $V$.

## 5.6 EFFECT OF LOCAL CLUSTERING CONSTRAINT

We propose to combine global and local clustering in GFC. To evaluate the contribution of the local clustering constraint, we conduct a series of experiments. Specifically, we disable the constraint by setting the regularization parameter $\beta = 0$. For comparison, we incorporate the constraint into FCM, which leads to an iterative solution similar to Eq. (15). Results in Table 5 show that introducing the local constraint yields a clear improvement in NMI, ACC, and ARI. The inclusion of local neighborhood information significantly strengthens the ability of fuzzy clustering to capture the underlying data structure, particularly in cases with complex local manifold patterns.

Table 5: Ablation study on the Local Clustering Constraint (LCC).

| Method | LCC | MNIST | | | USPS | | | CIFAR-20 | | |
|---|---|---|---|---|---|---|---|---|---|---|
| Metric | | NMI | ACC | ARI | NMI | ACC | ARI | NMI | ACC | ARI |
| FCM | ✗ | 48.9 | 54.6 | 37.0 | 63.9 | 62.3 | 53.2 | 47.1 | 42.7 | 25.9 |
| FCM | ✓ | 70.5 | 69.9 | 60.3 | 77.2 | 78.7 | 69.9 | 50.8 | 46.5 | 31.6 |
| GFC+SimCLR | ✗ | 75.0 | 82.4 | 71.8 | 73.1 | 77.4 | 66.3 | 47.6 | 46.9 | 31.7 |
| GFC+SimCLR | ✓ | 85.2 | 87.5 | 82.1 | 83.6 | 80.8 | 77.0 | 51.4 | 48.6 | 33.5 |

## 6 CONCLUSION

In summary, we present GFC, a new fuzzy clustering method for deep representation clustering. The key idea is to use the inner product of fuzzy probability vectors to capture global similarities and employ a local consistency constraint to model neighborhood relations. A theoretical formulation and an iterative optimization scheme are developed for GFC, along with a fast algorithm that reduces the complexity to linear time. Extensive experiments on multiple benchmarks demonstrate the effectiveness of our proposal.

## 7 ETHICS STATEMENT

This work does not raise any ethical concerns. All experiments are conducted on publicly available datasets, and no sensitive or personally identifiable information is involved.

## 8 REPRODUCIBILITY STATEMENT

We have made every effort to ensure the reproducibility of our work. The main paper provides detailed descriptions of the method architecture, along with the overall algorithmic framework and optimization procedure. To further support replication, we include comprehensive theoretical derivations, implementation details, and extended experimental results in the Appendix. Specifically, Sections A.2–A.3 present additional theoretical derivations; Sections A.5–A.6 and A.7 provide hyperparameter settings and pseudo-code, respectively. Extended experiments are reported in Sections A.8–A.11, covering parameter sensitivity analyses, ablation studies, additional clustering results on raw datasets, and execution time comparisons of GFC. We also release the source code in the supplementary submission to facilitate the replication of our results.

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

# A   APPENDIX

## A.1   THE USE OF LARGE LANGUAGE MODELS (LLMs)

We employed large language model to refine the grammar, phrasing, and overall readability of the abstract and introduction sections. The purpose of this usage was limited to improving language clarity and presentation. The models did not contribute to the generation of research ideas, methodology design, experimental analysis, or substantive content. All suggestions provided by the LLM were critically reviewed, verified, and, where appropriate, modified by the authors to ensure accuracy and alignment with the intended meaning.

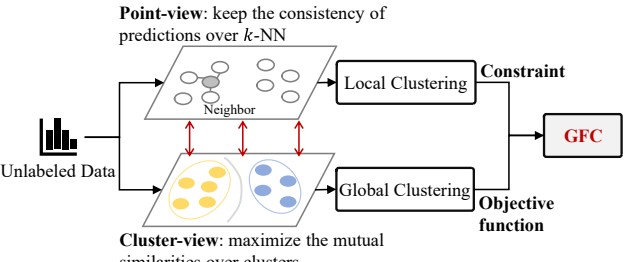

Figure 3: Overall research structure of GFC. GFC introduces an objective function for cluster-wise clustering and incorporates a consistency constraint for point-wise clustering. By combining global and local clustering approaches, GFC significantly enhances clustering performance.

## A.2   OPTIMIZATION DERIVATION

The overall structure of GFC is shown in Fig. 3. The objective function without local clustering constraint is simplified to

$$
\begin{aligned}
&\min_{\boldsymbol{P} \in \mathcal{P}} Tr(\boldsymbol{V}^T \boldsymbol{H} \boldsymbol{V} + \boldsymbol{P}^T \boldsymbol{H} \boldsymbol{P}^m - \alpha \boldsymbol{V}^T \hat{\boldsymbol{W}} \boldsymbol{P}^m) \\
&\Leftrightarrow \min_{\boldsymbol{P} \in \mathcal{P}} Tr((\boldsymbol{P}^T(\boldsymbol{1}_n \boldsymbol{1}_n^T - \boldsymbol{I}) - \alpha \boldsymbol{V}^T \hat{\boldsymbol{W}}) \boldsymbol{P}^m).
\end{aligned}
\tag{23}
$$

According to coordinate descent method, $\boldsymbol{p}_i$ is taken as the variable and the other elements as fixed constants, and the problem is decomposed into $n$ unconstrained sub-problems by Lagrangian multiplier method $(i = 1, \cdots, n)$ as

$$
\begin{aligned}
L_1(\boldsymbol{p}_i, a_i, \boldsymbol{b}_i) = &\sum_{l=1}^{c} \sum_{j \neq i} p_{i,l}^m (p_{j,l} - \alpha v_{j,l} \hat{w}_{i,j}) \\
&+ a_i (1 - \sum_{l=1}^{c} p_{i,l}) - \sum_{l=1}^{c} b_{i,l} p_{i,l},
\end{aligned}
\tag{24}
$$

where $a_i$ and $b_{i,l}$ are the Lagrangian multipliers respectively for the sum constraint and the non-negativity constraint on $\boldsymbol{p}_i$. The partial derivative of $L_1$ with respect to $p_{i,l}$ should be equal to zero at the minimum point, so we can obtain $(i = 1 \cdots, n; l = 1 \cdots, c)$

$$
\frac{\partial L_1}{\partial p_{i,l}} = m p_{i,l}^{m-1} \sum_{j \neq i} (p_{j,l} - \alpha v_{j,l} \hat{w}_{i,j}) - a_i - b_{i,l} = 0.
\tag{25}
$$

Then, we can get Karush-Kuhn-Tucker (K.K.T.) conditions Bertsekas (1997) $(i = 1, \cdots, n; l = 1, \cdots, c)$

$$
\begin{cases}
\frac{\partial L'}{\partial a_i} = 1 - \sum_{l=1}^{c} p_{i,l} = 0, \\
b_{i,l} p_{i,l} = 0, \\
b_{i,l} \geq 0.
\end{cases}
\tag{26}
$$

We define the fuzzy clustering score $s_{i,l}^p \in (0, +\infty)$ as $(i = 1 \cdots, n; l = 1 \cdots, c)$:

$$s_{i,l}^p = \sum_{j \neq i} (p_{j,l} - \alpha v_{j,l} \hat{w}_{i,j}), \tag{27}$$

and it has vector form

$$\boldsymbol{s}_i^p = [s_{i,1}^p, \cdots, s_{i,c}^p]^T = (\boldsymbol{P}^T \boldsymbol{1}_n - \boldsymbol{p}_i) - \alpha \boldsymbol{V}^T \hat{\boldsymbol{w}}_i, \tag{28}$$

where $\hat{\boldsymbol{w}}_i$ is the $i$ row of matrix $\hat{\boldsymbol{W}}$. The fuzzy clustering score indicates the score that $\boldsymbol{x}_i$ belongs to the cluster $k$.

Eq. (25) can be rewritten as $\frac{\partial L_1}{\partial p_{i,l}} = m p_{i,l}^{m-1} \boldsymbol{s}_i^p - a_i - b_{i,l} = 0$. Assume that $b_{i,l} > 0$, according to the conditional relation of Eq. (26), we have a solution: $p_{i,l} = 0$ and $a_i < 0$. Due to the properties of probability, there must exist $l^+$, $p_{i,l+} > 0$, $b_{i,l+} = 0$. Based on the above assumptions, we can get $\frac{\partial L_1}{\partial p_{i,l+}} = s_{i,l+}^p - a_i > 0$, which is contradictory to $\frac{\partial L_1}{\partial p_{i,l+}} = 0$. Therefore, we can draw a conclusion that $b_{i,l} = 0, p_{i,l} > 0, \forall l$, and Eq.(25) can be reformulated as $(i = 1, \cdots, n; l = 1 \cdots, c)$

$$p_{i,l} = \kappa \frac{a_i^{-1/(m-1)}}{(\sum_{j \neq i} 2p_{j,l}^m - \alpha v_{j,l} \hat{w}_{i,j})^{-1/(m-1)}}, \tag{29}$$

where $\kappa$ is a constant, $\kappa = m^{-1/(m-1)}$. Considering the sum constraint of probability, $\sum_{l=1}^c p_{i,l} = 1$, we have $(i = 1, \cdots, n)$

$$a_i^{-1/(m-1)} = \frac{1}{\kappa} \sum_{k=1}^c s_{i,k}^p {}^{-1/(m-1)}. \tag{30}$$

Substituting $a_i^{1/(m-1)}$ into Eq. (29), we can finally obtain following equivalence relation $(i = 1, \cdots, n; k = 1 \cdots, c$

$$p_{i,l}^* = \frac{s_{i,l}^p {}^{-1/(m-1)}}{\sum_{r=1}^c s_{i,r}^p {}^{-1/(m-1)}}. \tag{31}$$

From a mathematical perspective, Eq. (31) sharpens $\boldsymbol{s}_i^p$ by power operation and normalizes the clustering score to get the probabilistic output.

Then, we project the obtained $\boldsymbol{P}^*$ into the feasible region according to the local clustering constraint. The projected optimization problem is formulated as

$$\min_{\boldsymbol{P} \in \mathcal{P}} \|\boldsymbol{P} - \boldsymbol{P}^*\|_{\mathcal{F}}^2,$$
$$\text{s.t.} (\boldsymbol{I} - \boldsymbol{W} \boldsymbol{\Delta}^{-1}) \boldsymbol{P} = 0. \tag{32}$$

LCC is a slack constraint, which doesn't need to be strictly followed. We relax the constraint into the objective function and transform Eq. (32) into following problem

$$\min_{\boldsymbol{P} \in \mathcal{P}} \|\boldsymbol{P} - \boldsymbol{P}^*\|_{\mathcal{F}}^2 + \beta \|\boldsymbol{P} - \boldsymbol{W} \boldsymbol{\Delta}^{-1} \boldsymbol{P}\|_{\mathcal{F}}^2, \tag{33}$$

where $\beta > 0$ is the penalty hyperparameter. Eq. (33) is a label propagation problem Zhou et al. (2003) that limits the feasible domain to the probability space. We can then obtain the following Lagrange function $(i = 1, \cdots, n)$

$$L_2(\boldsymbol{p}_i, \boldsymbol{p}_i^*, c_i, \boldsymbol{d}_i) = \|\boldsymbol{p}_i - \boldsymbol{p}_i^*\|_2^2 + \beta \|\boldsymbol{p}_i - \frac{\sum_{j=1}^n w_{i,j} \boldsymbol{p}_j}{\sum_{j=1}^n w_{i,j}}\|_2^2$$
$$+ c_i (1 - \sum_{l=1}^c p_{i,l}) - \sum_{l=1}^c d_{i,l} p_{i,l}. \tag{34}$$

Using $\bar{p}_i$ to denote $\bar{p}_{i,l} = \frac{\sum_{j=1}^n w_{i,j} p_{i,l}}{\sum_{j=1}^n w_{i,j}}$, the partial derivative is equal to $(i = 1, \cdots, n)$

$$\frac{\partial L_2}{\partial p_{i,l}} = 2 * (p_{i,l} - p_{i,l}^*) + 2\beta (p_{i,l} - \bar{p}_{i,l}) - c_i - d_{i,l} = 0. \tag{35}$$

Afterwards, same as above solving process, taking K.K.T. condition into consideration, the solution is $(i = 1, \cdots, n; l = 1, \cdots, c)$

$$p_{i,l} = \frac{1}{1+\beta}p^*_{i,l} + \frac{\beta}{1+\beta}\bar{p}_{i,l}. \tag{36}$$

Therefore, the final optimal solution of Eq. (32) is

$$p_{i,l} = \frac{1}{1+\beta}\frac{s^p_{i,l}{}^{-1/(m-1)}}{\sum_{r=1}^c s^p_{i,r}{}^{-1/(m-1)}} + \frac{\beta}{1+\beta}\frac{\sum_{j=1}^n w_{i,j}p_{j,l}}{\sum_{j=1}^n w_{i,j}}. \tag{37}$$

The first term is the extreme point of the objective function without constraint. And the second term is the weighted average of the predictions of $k$-NN.

### A.3 Theoretical Analysis of Self-Constraint Terms

In this section, we analyze the roles of the self-constraint terms in GFC, which make the cluster output distribute uniformly. For hard label term $Tr(\boldsymbol{V}^T\boldsymbol{H}\boldsymbol{V})$, it can be reformulated as

$$\begin{aligned}
Tr(\boldsymbol{V}^T\boldsymbol{H}\boldsymbol{V}) &= Tr(\boldsymbol{V}^T\mathbf{1}_n^T\mathbf{1}_n\boldsymbol{V}) - Tr(\boldsymbol{V}^T\boldsymbol{V}) \\
&= \|\boldsymbol{V}^T\mathbf{1}_n\|_2^2 - \|\boldsymbol{V}\|_\mathcal{F}^2 \\
&= \sum_{l=1}^c(\sum_{i=1}^n v_{i,l})^2 - n.
\end{aligned} \tag{38}$$

According to Jensen inequality, we can obtain

$$\sum_{l=1}^c(\sum_{i=1}^n v_{i,l})^2 - n \geq \frac{1}{c}(\sum_{l=1}^c\sum_{i=1}^n v_{i,l})^2 - n = \frac{n^2 - cn}{c}. \tag{39}$$

To make the inequality hold with equality, we have $\sum_{i=1}^n v_{i,1} = \sum_{i=1}^n v_{i,2} = \cdots = \sum_{i=1}^n v_{i,c} = n/c$. For fuzzy label term $Tr(\boldsymbol{P}^T\boldsymbol{H}\boldsymbol{P}^m)$, according to the Eq. (31) we can get following relation $(i = 1, \cdots, n; l = 1, \cdots, c)$

$$p_{i,l} = \frac{\sum_{j\neq i} p_{j,l}{}^{-1/(m-1)}}{\sum_{r=1}^c \sum_{j\neq i} p_{j,r}{}^{-1/(m-1)}}. \tag{40}$$

Obviously, $p_{i,l} = 1/c, \forall i, l$ is the only minimum solution. Hence, when we minimize the objective function, the self-constrained regular terms tend to distribute the predictions evenly across each cluster.

### A.4 Details of Aggregation Acceleration Algorithms

The specific algorithm in the Section 4.5 is described in Algorithm 1.

---

**Algorithm 1:** Aggregation Acceleration Function: $F$

---
1 **Input:** $\tilde{\boldsymbol{P}}, \tilde{\boldsymbol{V}}, \mathcal{A}_i$
2 $\boldsymbol{p}' = \boldsymbol{v}' = 0$
3 **for** $j$ in $\mathcal{A}_i$ **do**
4 $\quad \boldsymbol{p}' += \boldsymbol{p}_j^m; \boldsymbol{v}' += \boldsymbol{v}_j$
5 **end**
6 $\boldsymbol{s}_i^p = \tilde{\boldsymbol{P}} - \alpha\boldsymbol{v}'; \boldsymbol{s}_i^v = \tilde{\boldsymbol{V}} - \alpha\boldsymbol{p}';$
7 **Output:** $\boldsymbol{s}_i^p, \boldsymbol{s}_i^v$

---

### A.5 Label Matrix Initialization

GFC initializes the hard assignment matrix $\boldsymbol{V}$ by K-means++ Arthur & Vassilvitskii (2006) algorithm, and initializes the matrix $\boldsymbol{P}$ according to a uniform distribution $(i = 1, \ldots, n; k = 1, \ldots, c)$

$$p_{i,k} = \frac{1}{c}. \tag{41}$$

### A.6 Avoid Negative Score

The clustering score $s_i^p$ is greater than zero in the derivation. However, to prevent algorithm failure due to $s_i^p$ becoming less than zero as a result of mini-batch clustering, we introduce an additional regularization formula for the calculation of $s_i^p$. The adjusted $s_i^p$ is defined as follows $(i = 1, \ldots, n; k = 1, \ldots, c)$

$$s_{i,k}^p = s_{i,k}^p - \min s_i^p + 1, \tag{42}$$

where $\min s_i^p$ is the minimum value of $s_i^p$. Eq. (42) ensures that the minimum value of $s_i^p$ is constrained, which facilitates the subsequent power operation.

### A.7 Algorithm Pseudo Code

Based on the above introduction, the entire Graph Fuzzy Clustering algorithm is summarized in Algorithm 2.

---

**Algorithm 2:** The Pseudo Code of GFC Algorithm

---

1 **Input:** Dataset $\mathcal{X}$, the number of cluster $c$, parameters $m$, $\alpha$, $\beta$, $n_b$
2 Calculate $\boldsymbol{W}$ and $\hat{\boldsymbol{W}}$ by Eq. (7) and Eq. (8);
3 Construct adjacent sample set $\mathcal{A} = \{\mathcal{A}_1, \cdots, \mathcal{A}_n\}$ based on $\hat{\boldsymbol{W}}$;
4 Initialize $\boldsymbol{V}$ randomly and set $p_{i,l} = 1/c, \forall i, l$;
5 **while** *not converge* **do**
6     Randomly Partition dataset $\mathcal{X} = \{\mathcal{X}_1, \cdots, \mathcal{X}_{[n/n_b]}\}$;
7     **for** $r = 1$ **to** $[n/n_b]$ **do**
8        $\tilde{\boldsymbol{P}} = \sum_{i=1}^n \boldsymbol{p}_i; \tilde{\boldsymbol{V}} = \sum_{i=1}^n \boldsymbol{v}_i$;
9        **for** $\boldsymbol{x}_i$ *in* $\mathcal{X}_r$ **do**
10           $\mathcal{A}_i^{'} = \mathcal{A}_i \cap \mathcal{X}_r$;
11           $\tilde{\boldsymbol{V}} - = \alpha \boldsymbol{v}_i; \tilde{\boldsymbol{P}} - = \alpha \boldsymbol{p}_i^m$;
12           $\boldsymbol{s}_i^p, \boldsymbol{s}_i^v = F(\tilde{\boldsymbol{P}}, \tilde{\boldsymbol{V}}, \mathcal{A}_i^{'})$ by Algorithm 1;
13           $\boldsymbol{s}_i^p = \boldsymbol{s}_i^p - \min \boldsymbol{s}_i^p + 1$
14           Update $\boldsymbol{p}_i$ by Eq. (15);
15           Update $\boldsymbol{v}_i$ by Eq. (18);
16           $\tilde{\boldsymbol{V}} + = \boldsymbol{v}_i; \tilde{\boldsymbol{P}} + = \boldsymbol{p}_i$;
17        **end**
18     **end**
19 **end**
20 **Output:** Fuzzy partition matrix $\boldsymbol{P}$

---

### A.8 More Results

For more experimental results, we evaluate our proposed GFC on six real-world datasets. To ensure the effectiveness of comparison methods, we adopt the default hyperparameters provided in the official implementations. For FCM, the fuzzy weighting exponent $m$ is fixed at 1.05. For graph-based clustering methods, including SC, and K-sums, $k$-NN graph is adopted. All algorithms are initialized randomly and run 50 times. The mean and variance of the clustering performance are reported to provide a comprehensive evaluation of each method.

The results are reported in the Table 6 where GFC consistently delivers superior performance and stability across most datasets. For instance, on MNIST, GFC achieves 0.853/0.877/0.821 (NMI/ACC/ARI), significantly outperforming FCM, which attains only 0.489/0.546/0.370. Moreover, with carefully selected parameters ($\theta = 3$, $k = 5$, and $n_b = 1000$), GFC reaches 0.914/0.966/0.926 (NMI/ACC/ARI), highlighting the potential of GFC under parameter tuning.

Table 6: Performance of Different Algorithms on Various Kinds of Data.

| Dataset | | PENDIGITS | | | ISOLET | | | MNIST | | |
|---|---|---|---|---|---|---|---|---|---|---|
| Metric | | NMI | ACC | ARI | NMI | ACC | ARI | NMI | ACC | ARI |
| KM | Avg | 0.680 | 0.699 | 0.553 | 0.726 | 0.553 | 0.486 | 0.497 | 0.546 | 0.376 |
| | (±Std) | (±0.011) | (±0.051) | (±0.034) | (±0.010) | (±0.027) | (±0.025) | (±0.014) | (±0.029) | (±0.020) |
| FCM | Avg | 0.684 | 0.683 | 0.561 | 0.725 | 0.535 | 0.479 | 0.489 | 0.546 | 0.370 |
| | (±Std) | (±0.013) | (±0.013) | (±0.038) | (±0.007) | (±0.024) | (±0.019) | (±0.011) | (±0.017) | (±0.013) |
| BIRCH | Avg | 0.727 | 0.680 | 0.552 | 0.757 | 0.600 | 0.541 | 0.701 | 0.645 | 0.551 |
| | (±Std) | (±0.000) | (±0.000) | (±0.000) | (±0.000) | (±0.000) | (±0.000) | (±0.000) | (±0.000) | (±0.000) |
| SC | Avg | 0.729 | 0.714 | 0.559 | 0.697 | 0.527 | 0.445 | 0.470 | 0.555 | 0.365 |
| | (±Std) | (±0.010) | (±0.014) | (±0.013) | (±0.007) | (±0.018) | (±0.015) | (±0.006) | (±0.016) | (±0.014) |
| FINCH | Avg | 0.725 | 0.627 | 0.501 | 0.723 | 0.445 | 0.418 | 0.726 | 0.623 | 0.548 |
| | (±Std) | (±0.000) | (±0.000) | (±0.000) | (±0.000) | (±0.000) | (±0.000) | (±0.000) | (±0.000) | (±0.000) |
| Self-CSC | Avg | 0.823 | 0.837 | 0.723 | 0.773 | 0.625 | 0.535 | 0.807 | 0.851 | 0.755 |
| | (±Std) | (±0.026) | (±0.037) | (±0.046) | (±0.007) | (±0.011) | (±0.012) | (±0.016) | (±0.036) | (±0.040) |
| K-sums | Avg | 0.657 | 0.726 | 0.571 | 0.728 | 0.615 | 0.524 | 0.509 | 0.585 | 0.420 |
| | (±Std) | (±0.000) | (±0.000) | (±0.000) | (±0.008) | (±0.016) | (±0.014) | (±0.010) | (±0.013) | (±0.014) |
| PAC | Avg | 0.702 | 0.773 | 0.616 | 0.734 | 0.617 | 0.530 | 0.495 | 0.607 | 0.409 |
| | (±Std) | (±0.004) | (±0.012) | (±0.004) | (±0.004) | (±0.001) | (±0.002) | (±0.005) | (±0.004) | (±0.005) |
| **GFC** | Avg | **0.837** | **0.869** | **0.770** | **0.773** | **0.629** | **0.575** | **0.853** | **0.876** | **0.823** |
| | (±Std) | (±0.016) | (±0.017) | (±0.023) | (±0.001) | (±0.001) | (±0.001) | (±0.001) | (±0.001) | (±0.001) |

| Dataset | | COIL-100 | | | USPS | | | EMNIST | | |
|---|---|---|---|---|---|---|---|---|---|---|
| Metric | | NMI | ACC | ARI | NMI | ACC | ARI | NMI | ACC | ARI |
| KM | Avg | 0.516 | 0.516 | 0.359 | 0.607 | 0.643 | 0.524 | 0.410 | 0.318 | 0.173 |
| | (±Std) | (±0.013) | (±0.032) | (±0.018) | (±0.004) | (±0.021) | (±0.009) | (±0.003) | (±0.006) | (±0.003) |
| FCM | Avg | 0.510 | 0.513 | 0.356 | 0.623 | 0.639 | 0.532 | 0.410 | 0.319 | 0.174 |
| | (±Std) | (±0.011) | (±0.035) | (±0.016) | (±0.013) | (±0.022) | (±0.022) | (±0.002) | (±0.004) | (±0.002) |
| BIRCH | Avg | 0.862 | 0.655 | 0.614 | 0.686 | 0.686 | 0.542 | 0.514 | 0.383 | 0.245 |
| | (±Std) | (±0.000) | (±0.000) | (±0.000) | (±0.000) | (±0.000) | (±0.000) | (±0.000) | (±0.000) | (±0.000) |
| SC | Avg | 0.791 | 0.517 | 0.373 | 0.489 | 0.561 | 0.295 | 0.274 | 0.269 | 0.050 |
| | (±Std) | (±0.000) | (±0.000) | (±0.000) | (±0.002) | (±0.002) | (±0.004) | (±0.021) | (±0.005) | (±0.006) |
| FINCH | Avg | 0.849 | 0.563 | 0.523 | 0.777 | 0.673 | 0.617 | 0.574 | 0.378 | 0.268 |
| | (±Std) | (±0.000) | (±0.000) | (±0.000) | (±0.000) | (±0.000) | (±0.000) | (±0.000) | (±0.000) | (±0.000) |
| Self-CSC | Avg | **0.882** | 0.715 | 0.602 | 0.726 | 0.754 | 0.614 | 0.586 | 0.497 | 0.322 |
| | (±Std) | (±0.005) | (±0.018) | (±0.014) | (±0.021) | (±0.034) | (±0.043) | (±0.003) | (±0.008) | (±0.006) |
| K-sums | Avg | 0.817 | 0.642 | 0.577 | 0.638 | 0.718 | 0.577 | 0.441 | 0.343 | 0.198 |
| | (±Std) | (±0.003) | (±0.007) | (±0.007) | (±0.005) | (±0.008) | (±0.006) | (±0.003) | (±0.007) | (±0.004) |
| PAC | Avg | 0.838 | 0.660 | 0.616 | 0.589 | 0.657 | 0.519 | 0.423 | 0.507 | 0.336 |
| | (±Std) | (±0.003) | (±0.008) | (±0.008) | (±0.000) | (±0.000) | (±0.000) | (±0.010) | (±0.012) | (±0.015) |
| **GFC** | Avg | 0.868 | **0.717** | **0.656** | **0.832** | **0.815** | **0.770** | **0.648** | **0.547** | **0.417** |
| | (±Std) | (±0.002) | (±0.008) | (±0.006) | (±0.002) | (±0.002) | (±0.003) | (±0.003) | (±0.009) | (±0.004) |

| Dataset | | CIFAR-10 (SimCLR) | | | CIFAR-20 (SimCLR) | | | STL-10 (SimCLR) | | |
|---|---|---|---|---|---|---|---|---|---|---|
| Metric | | NMI | ACC | ARI | NMI | ACC | ARI | NMI | ACC | ARI |
| KM | Avg | 0.750 | 0.770 | 0.639 | 0.471 | 0.427 | 0.261 | 0.651 | 0.668 | 0.503 |
| | (±Std) | (±0.022) | (±0.044) | (±0.037) | (±0.020) | (±0.027) | (±0.021) | (±0.028) | (±0.045) | (±0.041) |
| FCM | Avg | 0.754 | 0.784 | 0.651 | 0.464 | 0.427 | 0.261 | 0.682 | 0.683 | 0.511 |
| | (±Std) | (±0.014) | (±0.027) | (±0.026) | (±0.007) | (±0.013) | (±0.010) | (±0.029) | (±0.022) | (±0.032) |
| BIRCH | Avg | 0.658 | 0.629 | 0.544 | 0.414 | 0.363 | 0.244 | 0.582 | 0.650 | 0.484 |
| | (±Std) | (±0.000) | (±0.000) | (±0.000) | (±0.000) | (±0.000) | (±0.000) | (±0.000) | (±0.000) | (±0.000) |
| SC | Avg | 0.714 | 0.780 | 0.659 | 0.437 | 0.394 | 0.216 | 0.609 | 0.652 | 0.462 |
| | (±Std) | (±0.018) | (±0.043) | (±0.037) | (±0.007) | (±0.006) | (±0.007) | (±0.008) | (±0.022) | (±0.024) |
| FINCH | Avg | 0.728 | 0.592 | 0.545 | 0.478 | 0.350 | 0.204 | 0.592 | 0.433 | 0.384 |
| | (±Std) | (±0.000) | (±0.000) | (±0.000) | (±0.000) | (±0.000) | (±0.000) | (±0.000) | (±0.000) | (±0.000) |
| Self-CSC | Avg | 0.790 | 0.815 | 0.698 | 0.509 | 0.468 | 0.247 | 0.718 | 0.754 | 0.570 |
| | (±Std) | (±0.019) | (±0.022) | (±0.027) | (±0.012) | (±0.016) | (±0.022) | (±0.016) | (±0.034) | (±0.042) |
| K-sums | Avg | 0.714 | 0.780 | 0.659 | 0.491 | 0.471 | 0.323 | 0.629 | 0.697 | 0.547 |
| | (±Std) | (±0.016) | (±0.020) | (±0.029) | (±0.007) | (±0.018) | (±0.011) | (±0.028) | (±0.059) | (±0.047) |
| PAC | Avg | 0.759 | 0.871 | 0.763 | 0.464 | 0.439 | 0.295 | 0.662 | 0.753 | 0.599 |
| | (±Std) | (±0.002) | (±0.004) | (±0.004) | (±0.005) | (±0.006) | (±0.003)) | (±0.014) | (±0.029) | (±0.024) |
| **GFC** | Avg | **0.832** | **0.904** | **0.814** | **0.514** | **0.486** | **0.335** | **0.779** | **0.872** | **0.749** |
| | (±Std) | (±0.000) | (±0.000) | (±0.000) | (±0.008) | (±0.013) | (±0.010) | (±0.000) | (±0.000) | (±0.001) |

## A.9 PARAMETER SENSIBILITY ANALYSIS

We further investigate the parameter sensitivity of GFC by analyzing two key hyperparameters: the fuzzy weighting exponent $m$ and the number of nearest neighbors $k$ in the $k$-NN graph. These parameters respectively control the degree of fuzziness in cluster assignments and the scope of local neighborhood influence, both of which play critical roles in shaping the clustering behavior. To ensure fair evaluation, we vary one parameter at a time while fixing the others to their default settings.

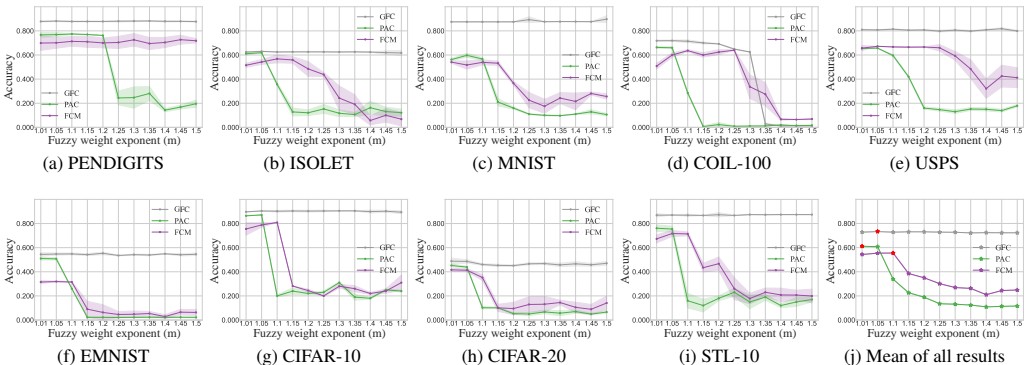

Figure 4: Accuracy comparison of GFC, PAC, and FCM with different weighting exponent $m$.

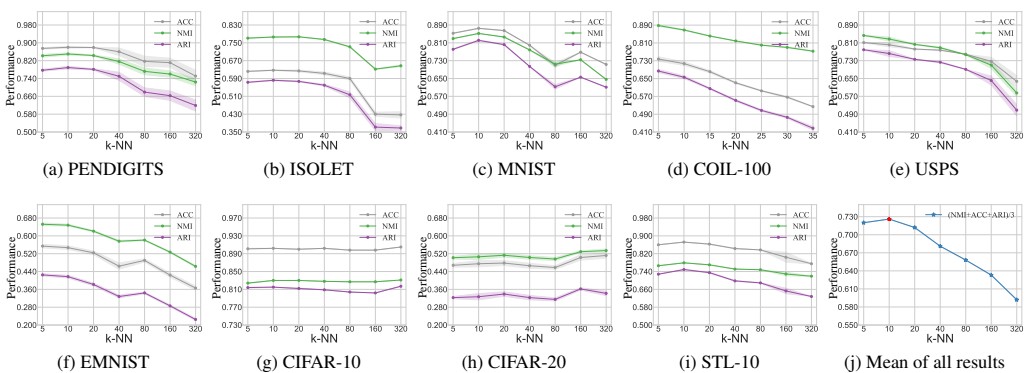

Figure 5: Performance of GFC with different $k$ in $k$-NN.

The fuzzy weighting exponent $m$ determines the sharpness of membership probabilities and directly influences the confidence of cluster assignments. We evaluate the impact of $m$ by comparing GFC, PAC, and FCM within the range $(1, 1.5]$ with a step size of $0.05$, and the results are reported in Fig. 4. Overall, GFC demonstrates strong robustness to $m$, maintaining stable performance across a wide range of values. This suggests that the integration of local constraints alleviates the sensitivity commonly observed in fuzzy clustering algorithms. Nevertheless, on COIL-100, where the number of clusters is relatively large, higher values of $m$ lead to overly fuzzy predictions and degraded performance. A practical guideline is thus to select smaller $m$ values when clustering tasks involve a large number of clusters.

We examine the sensitivity to the $k$-NN parameter by varying $k \in 5, 10, 20, 40, 80, 160, 320$. The results, summarized in Fig. 5, show a clear trend: as $k$ increases, clustering accuracy tends to decrease. This is because enlarging the neighborhood dilutes local structural information, making the constraint approximate a global one. Furthermore, larger $k$ values bring higher computational cost without significant benefit. Based on the average results across all datasets, we find that $k = 10$ provides the best trade-off between clustering quality and efficiency.

We test the influence of $\alpha$ and $\beta$ and the results shown in Fig. 6, respectively. From experiments we observe that $\alpha > 0.5$ provides a good balance between clustering and self-constrained terms, and $\beta$ controls the strength of local clustering, and its effect improves as $\frac{\beta}{1+\beta}$ approaches 1.

In summary, these experiments confirm that GFC is robust to the choice of $m$ and performs best with moderate $k$ values, offering practical guidance for parameter selection in real-world applications.

## A.10 IMPACT OF LOCAL CLUSTERING CONSTRAINTS ON MINI-BATCH CLUSTERING

The experimental results in Table 5 highlight the significant accuracy gains achieved by incorporating the Local Consistency Constraint (LCC). By leveraging local neighborhood information, LCC

enhances the capacity of fuzzy clustering to capture fine-grained structures, especially in datasets with complex or highly localized patterns. Moreover, as illustrated in Fig. 7, LCC effectively alleviates the performance degradation typically observed in mini-batch global clustering, where the absence of local context often leads to suboptimal solutions.

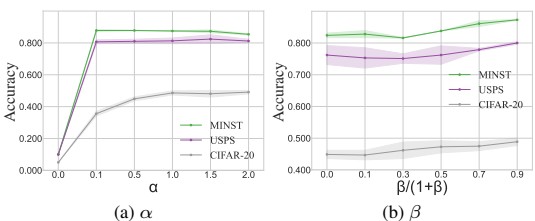

Figure 6: Performance of GFC with different $\alpha$ and $\beta$.

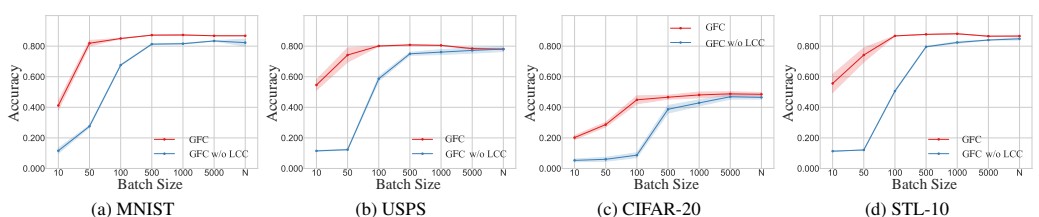

Figure 7: Performance of GFC and GFC w/o LCC with different batch size.

## A.11    EXECUTION TIME

The theoretical time complexity of GFC algorithm is linear. To evaluate its practical execution time, we sample subsets of varying sizes from the MNIST dataset. We record the actual running time of the GFC optimization process, scaling with increasing sub-dataset size. For comparison, we also report the running times of several other algorithms, including PAC, Self-CSC, K-sums, and FCM. The algorithms PAC, GFC, and FCM are implemented in Python, Self-CSC in Matlab, and K-sums in C++. The reported running time reflects the entire execution time, including the computation of pairwise distances, the construction of the $k$-NN graph, and the optimization iterations. All experiments are conducted on a personal computer equipped with an AMD Ryzen 9 7900X 12-Core Processor (4.70 GHz) and 64 GB RAM, using the official codes for each algorithm. The results are shown in Fig. 8. GFC demonstrates exceptional time efficiency, closely matching the performance of FCM and K-sums, despite Python being slower in handling loops compared to C++ and Matlab.

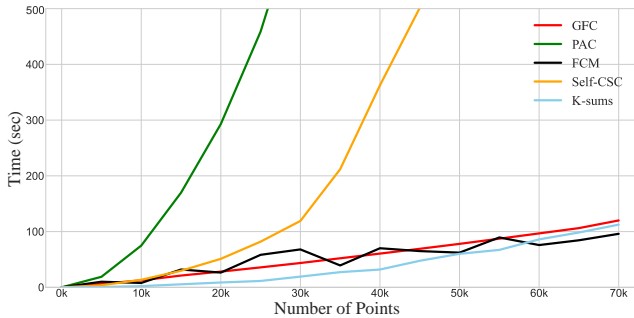

Figure 8: Running times for GFC and other algorithms on MNIST subsets.

