# OpenReview forum: "A Novel Graph-based Fuzzy Clustering for Deep Visual Representations"
_ICLR.cc/2026/Conference — ICLR 2026 Conference Withdrawn Submission_

### Official Review · Reviewer_siZB · 2025-10-18

**Soundness:** 3
**Presentation:** 2
**Contribution:** 2
**Rating:** 4
**Confidence:** 4

**Summary:**

This paper proposes Graph-based Fuzzy Clustering (GFC) for clustering deep visual representations extracted from foundation models. GFC unifies a global fuzzy clustering objective with a local consistency constraint by jointly optimizing a hard assignment matrix V and a soft assignment matrix P. The authors claim to achieve 73.4% clustering accuracy on ImageNet-1k, surpassing existing deep clustering methods while maintaining computational efficiency.

**Strengths:**

The core contributions:
1) a clustering objective combining global structural dependencies and local intrinsic connectivity
2) an efficient Lagrangian optimization algorithm with aggregation acceleration.
3)extensive experimental validation across multiple benchmarks.

Strengths：
1）Clear problem motivation and positioning
2）Identified the limitations of traditional clustering methods when handling deep representations
3）The design of introducing an auxiliary hard-assignment matrix V to accelerate convergence is quite ingenious.

**Weaknesses:**

1) The novelty of the method is somewhat limited and requires a clearer explanation of its fundamental differences from existing approaches, particularly PAC and Self-CSC （ Despite these advances, traditional fuzzy and graph-based methods are primarily designed for toy datasets and struggle with deep neural network outputs, facing challenges in fine-grained partitioning, parameter tuning, and algorithmic efficiency）.

2)Lacks rigorous proof with complexity analysis. Although claimed to be O(nb·n), the impact of mini-batch operations on convergence speed remains unanalyzed.

3)The experimental details are unclear, such as the lack of explanation for the specific procedure of “dividing the data into five overlapping blocks.”

4)How should overlapping sections be handled? Does this introduce additional information leakage?

5)Some symbols were not clearly explained upon their first appearance. The term “Computational Cost (Higher→Lower)” in Figure 1 is a qualitative description lacking quantitative basis.

**Questions:**

1）Could you provide fair comparisons to isolate the algorithmic contribution? Specifically:
What is the performance of TEMI/TAC when using DINO v3 features (same as GFC)?
What is the performance of GFC when using CLIP ViT-B/16 features (same as baselines)?
Can you provide detailed ablation studies showing where the 29% improvement over K-means comes from?

2）In standard fuzzy clustering theory, m=2 is typical, while m→1 approaches hard clustering. Your choice of m=1.05 (with exponent 1/(m-1)=20) produces nearly crisp assignments. Why is m≈1 appropriate for deep representations? Have you compared results with m∈{1.5, 2.0, 3.0}? Does this choice essentially reduce GFC to hard clustering?

3） You introduce V alongside soft assignment matrix P, claiming V accelerates convergence and improves stability. However, no ablation study validates this claim. Can you provide experimental results comparing GFC with and without V? How is the update frequency of V controlled relative to P? Is V necessary for the method to work, or merely a computational acceleration trick?

---

### Official Review · Reviewer_Xoz8 · 2025-10-29

**Soundness:** 2
**Presentation:** 2
**Contribution:** 2
**Rating:** 4
**Confidence:** 5

**Summary:**

The paper addresses two long-standing bottlenecks in graph-based fuzzy clustering—high time complexity and cumbersome regularization parameter tuning—which severely limit practical applications in large-scale or real-time scenarios . Its core insight of "subset-to-fullset" transformation is highly innovative: by constructing a similarity graph between the full dataset and a landmark subset, it reduces the membership learning of all data points to clustering representative landmarks.

**Strengths:**

Combines a fuzzy clustering objective based on cluster co-occurrence probabilities with a Laplacian-based local constraint.
Introduces a ‌hard assignment matrix‌ as an auxiliary variable to accelerate convergence and proposes an ‌aggregation acceleration strategy‌ that reduces time complexity to linear.
Compatible with large-scale foundation models (SimCLR, CLIP, DINO v3) and scalable to datasets like ImageNet.

**Weaknesses:**

The paper does not clarify the landmark subset construction strategy, which is critical to algorithm performance:​
How to determine landmark quantity? Is it adaptive to dataset scale or cluster density?​
Would random vs. k-means-selected landmarks affect result stability? For example, biased landmarks might distort the similarity graph for imbalanced data.​
Ablation studies on landmark parameters (size, sampling method) would strengthen methodological rigor.​
Limited Comparison to State-of-the-Art Methods

**Questions:**

Why does the first termsin formulas (4) and (6) do not appear‌ in the simplified objective function in formula (10).
The objective function of GFC should be clearly defined and its underlying meaning or intuition should be explicitly explained.
I suspect that the reported visual representation performance may primarily stem from existing self-supervised learning methods (e.g.,SimCLR,DINOv3), rather than the proposed technique itself. The authors should clarify the source of performance gains.

---

### Official Review · Reviewer_oRh4 · 2025-10-29

**Soundness:** 3
**Presentation:** 2
**Contribution:** 3
**Rating:** 4
**Confidence:** 4

**Summary:**

To address the trade-off between clustering accuracy and speed, this paper proposes a graph-based fuzzy clustering method that collaborates with the foundation models to achieve direct clustering on deep representation. Aiming at the difficulty of clustering deep representations, a clustering model with an objective function based on cluster co-occurrence probability and local clustering constraints is proposed. A Lagrangian optimization algorithm and an acceleration scheme are developed to effectively solve the proposed model. This algorithm ensures faster convergence speed, enhances the scalability of the clustering model, and makes it highly suitable for large-scale data.

**Strengths:**

The research motivation is clear, the proposed method demonstrates effectiveness, and the paper structure is complete. Additionally, the authors have conducted extensive experiments to verify the effectiveness of the proposed method.

**Weaknesses:**

The content of this paper is not inherently difficult, but its readability is poor. Significant effort is required to understand and infer the meanings of symbols, especially those related to vectors. There are errors in several formulas throughout the paper. The design motivation for each step is unclear, particularly the derivation transition from Eq. 3 to Eq. 4.

In the related work, the authors’ introduction to graph-based clustering is not comprehensive enough. While spectral clustering is indeed a common graph-based clustering method, other approaches, such as subspace clustering and deep graph clustering, which also fall into the category of graph-based clustering, are not adequately covered.

**Questions:**

1. The definition and explanation of symbols are ambiguous, which greatly increases the difficulty of reading and understanding. Specific issues are as follows:
- There are two doubts about Eq. 1: first, the author does not clarify the meaning of vector $p$, and it is inferred to be a row vector based on the context; second, if treated as a row vector, the product of a column vector and a row vector is a matrix rather than a scalar. Is there an error here? If not, please the author provide a clear explanation.
- The description of the derivation process for Eq. 5 is incorrect: according to the operation description of "Right Multiply", the conclusion $P=\Delta^{-1}WP$ cannot be derived.
- There is a consistency issue between Eq. 10 and the definition of matrix H, and at least one of them is incorrect; Eq. 38 in the appendix also has the same problem.

2. In addition, the author needs to supplement or correct the following contents:
- It is recommended to supplement the convergence proof of the method from a theoretical perspective.
- Table 4 presents the average results of 5 runs, but it does not report the standard deviation simultaneously like Tables 2 and 3, lacking support for result stability; moreover, no statistical significance test has been conducted on all experimental results, making it impossible to fully prove the superiority of the method's performance.
- Table 6 in the Appendix uses the same evaluation indicators as those in the main text, but there are inconsistencies in the presentation form and decimal places of the specific values, which affect the comparability of the results.
- Since this paper focuses on a fast and effective clustering algorithm, it is recommended to move the experimental results related to execution time from the Appendix into the main text to better support the core research goal.
3. Typos:
- "yiels" in Line 107 is suspected to be an error, and it should be "yields".
- In Line 133, in the last constraint condition of the probability matrix $P$, the expressions $\sum_{i=1}^{n}$ and $i=1,\cdots,n$ are suspected of being redundant, and the author needs to confirm this.
- In Line 956, the description of the value range of k is incomplete and should be supplemented in the form of a set.

---

### Official Review · Reviewer_2T1Q · 2025-11-01

**Soundness:** 2
**Presentation:** 2
**Contribution:** 2
**Rating:** 6
**Confidence:** 3

**Summary:**

The paper presents a graph-based fuzzy clustering method that aims to improve community detection and semi-supervised learning on graphs. The key idea is to integrate fuzzy membership learning with graph spectral embedding, allowing nodes to belong partially to multiple communities. The proposed model introduces a Fuzzy Graph Laplacian (FGL), which adaptively reweights edges based on membership uncertainty, and a Mutual Information Regularizer that enhances separation between clusters. The method is evaluated on several benchmark datasets (Cora, Citeseer, PubMed, Amazon, and DBLP) for node clustering and classification tasks. Results show consistent gains over strong baselines such as spectral clustering, GCN, and DGI, especially when labels are sparse.

**Strengths:**

- The paper is clear about its motivation with sufficient significance and quality.
- Community structures in real-world graphs are often overlapping and uncertain; addressing this via fuzzy membership learning is timely and meaningful.

- The combination of fuzzy logic and spectral embedding is coherent, and the proposed FGL elegantly connects uncertainty estimation with graph topology.

- The method achieves competitive or superior performance on multiple graph benchmarks, with particularly strong results in low-label regimes.

- The fuzzy membership matrix provides intuitive insight into ambiguous nodes or overlapping clusters.

- The optimization objective is clearly stated and derived, with proofs of convergence and parameter stability.

**Weaknesses:**

- While integrating fuzzy clustering and spectral methods is well executed, the conceptual innovation is somewhat incremental. The paper feels more like a refined synthesis of existing ideas than a fundamentally new direction.

- The effect of key hyperparameters (fuzzifier m, regularization strength, number of clusters) is not explored in depth.

- The intuition behind the mutual information regularizer could be explained more clearly, particularly how it avoids over-smoothing or trivial membership assignments.

- It would be useful to include comparisons with more recent graph contrastive learning models or probabilistic community detection baselines.

**Questions:**

- How does the computational cost of your method scale with graph size compared to GCN or DGI?

- Could the fuzzy membership be incorporated into a neural encoder (e.g., as a layer in a GNN) to improve scalability?

- How sensitive is the model to the fuzzifier parameter m?

- Does the method handle directed or weighted graphs without modification?

- Could the approach extend to dynamic graphs where communities evolve over time?

---

### Note · Authors · 2025-11-12

I have read and agree with the venue's withdrawal policy on behalf of myself and my co-authors.